# Inelastic Deformation of Coronary Stents: Two-Level Model

**DOI:** 10.3390/ma15196948

**Published:** 2022-10-07

**Authors:** Pavel S. Volegov, Nikita A. Knyazev, Roman M. Gerasimov, Vadim V. Silberschmidt

**Affiliations:** 1Department of Mathematical Modeling of Systems and Processes, Perm National Research Polytechnic University, 614990 Perm, Russia; 2Wolfson School of Mechanical, Electrical and Manufacturing Engineering, Loughborough University, Leicestershire LE11 3TU, UK

**Keywords:** coronary stents, FEM, multilevel modelling, inelastic deformation, biomaterials, strain

## Abstract

This study describes the internal structure of materials used to produce medical stents. A two-level elastoviscoplastic mathematical model, which sets the parameters and describes the processes at the grain level, was developed and numerically implemented. A separate study was conducted to identify the most dangerous deformation modes in the balloon-expandable stent placement using the finite-element method in COMSOL Multiphysics. As a result, the challenging strain state type required for setting the kinematic loading on a representative macrovolume in the two-level model was obtained. A yield surface for different deformation paths in the principal deformation space for stainless steel AISI 316L was obtained and the effect of grain size on the deformation behavior of this material was explored using the developed model.

## 1. Introduction

Coronary stents, made of metal alloys and biopolymers, are used to expand blocked blood vessels and maintain sufficient blood flow in the human body, [1]. The use of coronary stents ensures the mechanical strength and integrity of structures [2]. The choice of stent material is highly dependent on its physical and mechanical characteristics, biocompatibility, rate of stent degradation in the body, etc. For example, to make self-expanding stents, nickel-titanium alloys are used because they have excellent flexibility, strength, biocompatibility, superelastic behavior, and shape memory effect [3,4,5,6]. However, alternative titanium and niobium alloys were used in some instances [7,8,9] due to concerns about the diffusion of nickel compounds from the stent surface and the inflammatory response of body cells to nickel ions. Nickel-free alloys have recently become more favorable in the medical industry. Cobalt-chrome (Co-Cr) alloys with other alloying elements (W, Mo, etc.) are among the most common material groups for biomedical purposes [10,11,12,13]. They have greater corrosion and wear-resistance compared with Al-Mg or Ti alloys. However, it is worth noting that the properties of biomedical material devices not only depend on the chemical composition of materials, but also on the microstructure morphology obtained in the mechanical, thermal, and thermomechanical treatment processes.

Balloon-expandable stents, unlike the self-expanding stents, experience large plastic deformations resulting in their self-hardening [14,15]. Such stents are usually made from stainless steel or cobalt-chrome-based alloys [16,17]. Some authors [17,18,19,20,21] suggest using pure iron, magnesium, and zinc alloys, as well as poly-L-lactic acid biopolymer for stents, as these materials are able to degrade and resorb in the body after implantation. The focus of this paper is on balloon-expandable stents, as they are more commercially available. Stents are classified according to their structure, as helical, slotted tube, or modular [22,23,24]. A helical stent is a wire coil that forms a stent frame. Slotted tube stents are produced from small tubes with an etched core, using laser-etching techniques. The frame of modular stents is formed by a system of hoops. The geometry of cell structures in modular stents is the main design factor, defined by the number and layout of hoop connectors. It is desirable to reduce the number of connectors in a stent to increase its maximum flexure and reduce challenges caused by transportation. It has recently become apparent that longer, thinner, and more flexible stents can be less stable along their longitudinal axis. These stents can be compressed or deformed along a device, producing a so-called accordion effect or longitudinal deformation of the stent [25,26].

Grain boundaries play a significant role in the plastic-deformation and fracture processes of polycrystalline materials. The grain (or intergrain) boundary is the interface between two crystals with different crystallographic orientations. The intergrain boundary layer has a pronounced defect structure, with a characteristic thickness of approximately two or three interatomic distances in pure metals [27]. High stresses and deformations are often localized near such boundaries, significantly disturbing crystal lattices in grain-boundary regions [28]. On the one hand, such strain localization is explained by the influence of dislocation pile-ups, formed from successive deceleration of single dislocations at the barriers (including grain boundaries) during plastic deformation. On the other hand, grain boundaries with a defect structure and increased dislocation density, compared with the interior of the grains, generate lattice distortions in the boundary regions. Thus, intergrain boundaries have a direct impact on the mechanism of grain-boundary sliding (GBS), as well as accommodation mechanisms, preventing material discontinuity in the case of a single GBS mechanism [29]. In addition, grain boundaries restrain dislocation movements and impose limitations on plastic deformation of a grain surrounded by differently oriented adjacent grains [30]. Researchers still have different views on lattice dislocations passing through the grain boundary [31,32]. Many studies describing the initiation and accumulation of damage in metals [33,34,35] have used a model of a planar dislocation pile-up at the grain boundary as a fracture mechanism. Indeed, long-range stress fields of grain-boundary and lattice dislocations in the pile-ups of a grain can break the interatomic bonds with a subsequent fracture in the adjacent grains.

The fact that the grain boundaries are linked to GBS, and accommodation mechanisms defines the parameters of our model that characterizes the influence of the boundaries on the material’s deformation character. The impact of intergrain boundaries is apparent in the Hall–Petch law [36], which relates the yield strength of the material to an average grain size in a polycrystal. Namely, as the volume fraction of the grain-boundary region increases in the material [37] with a grain-size decrease, it raises the deformation resistance and yield strength (or material hardening).

In models of inelastic deformation of polycrystals, hardening laws are usually formulated as evolution equations for critical shear stresses of the dislocation motion [38,39,40,41]. The critical stresses are often increased due to different kinds of barriers, which include grain boundaries. Thus, when describing grain-boundary hardening, it is important to consider scenarios of slip transmission for dislocations across the boundary with a subsequent continuation of plastic deformation in the adjacent grains. As a result of this absorption of lattice dislocations, misfit dislocations are formed in the grain boundaries, with their Burgers vector equal to a difference between the Burgers vectors of dislocations entering the boundary and emerging from it. Grain-boundary hardening depends on the sizes and orientations of grains, as well as their relative location. Therefore, to describe this behavior, the patterns and orientation distributions of the real grain structure in metals need to be understood.

A literature review on the grain structure used for the modeling of metals demonstrated that the grain sizes in a polycrystal follow the lognormal distribution law, which is consistent with experimental data [42,43,44,45,46]. It is noted in [43,44] that a significant change in the grain structure and parameters of the lognormal distribution law occurs during the heat treatment of materials. For instance, an increase in the annealing temperature and time leads to grain growth in the polycrystal. In 316L stainless steels, an average grain size usually varies from 6 to 80 μm [10,47,48], with a standard deviation of 0.35 [46].

The features of the grain structure in coronary stents were studied with images of a 316L stainless steel stent strut obtained with scanning electron microscopy [49,50] (Figure 1). These images indicate a rather small length of the stent strut in two directions. Based on the previously obtained average grain size in the 316L steel, it can be concluded that the structure in width and thickness consists of approximately 1–13 grains.

Texture formation is another problem for grain-structure modeling of such components. As noted in [49,50], a texture mainly occurs due to the extrusion of the initial tubular samples; however, final annealing significantly reduces textures in the stent material. Thus, many elastoplastic and elastoviscoplastic models of the metals use a uniform distribution of grain orientations [48,49]. This type of distribution was also used in this work.

Fractures are the most common problem related to the use of stents, as well as the arterial damage from direct contact between the arteries and stent. Various sources report that stent rupture occurs in 1–18% of cases, which is quite a serious challenge for their wide applications [50,51,52]. In most cases, the stents are made of metals and alloys, the mechanical strength of which is determined by their structure. One of the main reasons for stent damage is the pronounced anisotropy of properties caused by the small thickness of the stent tube, some 10–20 grains through in thickness. Similar cases are widely known in the mechanics of solids as statistical size effects (SSE), which in most cases has a negative impact on mechanical characteristics of struts exposed to large plastic deformation. Current design is focused on stents with thinner struts, which requires an account for SSE [10]. Obtaining morphological properties of a material is often a complex problem, as it is necessary to carry out several studies of the structure, up to its complete destruction. The influence of grain size and characteristic orientation map are hypothesized to be the key aspects to consider in the model. The statistical nature of the model is one of the requirements, as such parameters can differ even under the same manufacturing conditions and chemical compositions of a material. To control the properties of products, it was necessary to develop an accurate mathematical model describing the internal structure of the material. Thus, an explicit consideration of the material morphology that takes the features of its structure into account is the most important aspect of stent modeling.

## 2. Materials and Methods

The properties and behavior of a material at the sample level (macrolevel) significantly depend on the evolution of its meso- and microstructure and should be properly considered in a model of the inelastic deformation processes of polycrystals [53]. So, the physical approach based on the direct consideration of the mechanisms and processes at the meso- and microlevel has recently become popular [29]. To describe the evolution of the underpinning mechanisms, researchers have introduced parameters reflecting the state and evolution of meso- and microstructure and formulate evolution equations for them [38,54,55].

In this paper, a two-level (macro-meso) elastoviscoplastic model of inelastic deformation of a polycrystalline aggregate was developed. A macrolevel element is the representative volume of a polycrystal consisting of many mesolevel elements, i.e., grains (crystallites). Apparently, a representative volume is the minimum volume of a material with enough elements responsible for the considered process mechanisms for the statistical description of its state [56].

It is worth noting that the additivity hypothesis of the elastic and inelastic components of the strain-rate measure at the meso- and macrolevel is accepted in most elastoviscoplastic models [29,53]:(1)$$\zeta =\xi^{e}+\xi^{in},$$
(2)$$Z=Z^{e}+Z^{in},$$
where ζ is the strain-rate measure at the mesolevel and $\xi^{e}$ and $\xi^{in}$ are the elastic and inelastic components of the measure at the mesolevel, respectively. Z is the strain-rate measure at the macrolevel and $Z^{e}$ and $Z^{in}$ are the elastic and inelastic components of the measure at the macrolevel, respectively. To transfer the effect of the macrolevel to the mesolevel, the extended Voigt hypothesis is used:(3)ζ=Z.

Hooke’s law in the rate form serves as the constitutive relation at the mesolevel:(4)$$\dot{\sigma }=п:(\zeta -\zeta^{in}),$$
where $\dot{\sigma }$ is the material derivative of the Cauchy stress tensor, п is the tensor of the crystallite’s elastic properties, and $\zeta =\hat{\nabla }v^{T}$ is the mesoscale strain-rate measure, coinciding with the transposed velocity gradient. All values in Equation (4) were determined in the crystallographic coordinate system (CCS) of the crystallite.

The main mechanism of inelastic deformation in the material is the intragrain dislocation slip along the most closely packed planes and directions. The combination of the plane and direction forms the slip system (SS), which is characterized by the Burgers vector **b** in the slip direction and the slip plane’s normal **n**. Assuming a uniform dislocation distribution in grains, the inelastic component of the strain rate measure can be calculated [29,53]:(5)$$\zeta^{in}=\sum_{k=1}^{K}{\dot{\gamma }}^{\left(k\right)}n^{\left(k\right)}b^{\left(k\right)},$$
where *K* is the total number of the slip systems and ${\dot{\gamma }}_{0}$ is the shear rate for the *k*-th slip system.

To establish the shear rates in the slip systems, the following power law can be applied:(6)$${\dot{\gamma }}^{\left(k\right)}={\dot{\gamma }}_{0}{\left|\frac{\tau^{\left(k\right)}}{\tau_{c}^{\left(k\right)}}\right|}^{\frac{1}{m}}H(\tau^{\left(k\right)}-\tau_{c}^{\left(k\right)}),$$
where $\tau_{c}^{\left(k\right)}$ is the critical shear stress for the *k*-th slip system, ${\dot{\gamma }}_{0}$ is the characteristic shear rate in the case when the acting tangential stress is equal to the critical stress in the slip system, and *m* is the constant of the rate sensitivity of the material. $\tau^{\left(k\right)}$ is the acting tangential stress for the *k*-th slip system, defined as
(7)$$\tau^{\left(k\right)}=\sigma :n^{\left(k\right)}b^{\left(k\right)}.$$

The transition between the scale levels is performed by employing an averaging procedure for the mesolevel element’s characteristics. The constitutive relation at the macrolevel is Hooke’s law in the rate form:(8)$$\dot{\Sigma }=П:(Z-Z^{in}),$$
where $\dot{\Sigma }$ is the material derivative of the Cauchy stress tensor at the macrolevel, and П is the macroscale tensor of the crystallite elastic properties. All the values in Equation (8) were determined in the laboratory coordinate system (LCS).

Thus, the systems of equations at the meso- and macrolevel have the form (*N* is the grain number):

Mesolevel:$$\{\begin{array}{c} \dot{\sigma }=п:(\zeta -\zeta^{in}),\\ \zeta^{in}=\sum_{k=1}^{K}{\dot{\gamma }}^{\left(k\right)}n^{\left(k\right)}b^{\left(k\right)},\\ {\dot{\gamma }}^{\left(k\right)}={\dot{\gamma }}_{0}{\left|\frac{\tau^{\left(k\right)}}{\tau_{c}^{\left(k\right)}}\right|}^{\frac{1}{m}}H(\tau^{\left(k\right)}-\tau_{c}^{\left(k\right)}),\\ \tau^{\left(k\right)}=\sigma :n^{\left(k\right)}b^{\left(k\right)},\\ {\dot{\tau }}_{c}^{\left(k\right)}=f({\dot{\gamma }}^{\left(k\right)},\gamma^{\left(k\right)},\ldots ). \end{array}$$

Macrolevel:$$\left\{ \begin{array}{l} \dot{\Sigma }=П:(Z-Z^{in}),\\ П=\langle {п}_{\left(i\right)}\rangle ,\\ Z^{in}=\langle \xi_{\left(i\right)}^{in}\rangle ,\\ i=1,\ldots ,N. \end{array}\right.$$

The evolution of the critical shear stresses for each SS is realized due to mechanisms of the intragrain and grain-boundary hardening:(9)$${\dot{\tau }}_{c}^{\left(k\right)}=f^{\left(k\right)}+f_{\mathrm{GBH}}^{\left(k\right)}.$$

The intragrain hardening mechanism considers the dislocation interactions at different slip systems. The relation for the critical stress rate has a classical form [57,58]:(10)$$\left\{ \begin{array}{l} f^{\left(k\right)}=\sum_{l=1}^{24}h^{\left(kl\right)}\;{\dot{\gamma }}^{\left(l\right)},\;k=1,\ldots ,24,\\ h^{\left(kl\right)}=\left[q_{lat}+(1-q_{lat})\delta^{kl}\right]\;h^{\left(l\right)},\\ h^{\left(l\right)}=h_{0}{\left|1-\frac{\tau_{c}^{\left(l\right)}}{\tau_{sat}}\right|}^{a}, \end{array}\right.$$
where $q_{lat}$ is the latent-hardening parameter, $\delta^{kl}$ is the Kronecker delta, $h^{\left(kl\right)}$ is the hardening-modulus matrix, $\tau_{sat}$ is the reference stress, at which plastic flow initiates, ${\dot{\gamma }}^{\left(l\right)}$ is the shear rate for the *l*-th slip system, and *h*_0_ and *a* are the hardening-law parameters.

The rate of increase in the critical shear stresses due to the grain-boundary hardening mechanism is defined as follows [39,41]:(11)$$f_{\mathrm{GBS}}^{\left(k\right)}=\eta \;{\dot{\gamma }}^{\left(k\right)}\gamma^{\left(k\right)}\sum_{i=1}^{P}\frac{S_{i}}{V}\xi_{i}^{\left(k\right)},$$
where η is the hardening-law parameter, V is the grain volume, $S_{i}$ is the contact area between the current and adjacent grain, $\xi_{i}^{\left(k\right)}$ is the misorientation measure for these grains, ${\dot{\gamma }}^{\left(k\right)}$ and $\gamma^{\left(k\right)}$ are the shear rate and the accumulated shear for the *k*-th SS, respectively, and *P* is the number of regions approximating the grain boundaries. The misorientation measure $\xi_{i}^{\left(k\right)}$ has the form:(12)$$\xi_{i}^{\left(k\right)}=\underset{l=1,24}{\min}\left\{\left|b^{l}-b^{k}\right|\cdot N^{i}\right\},$$
where $b^{k}$ and $b^{l}$ are the dislocation Burgers vectors passing from the slip systems of the current and adjacent grains, respectively, $N^{i}$ is the normal to the boundary region.

## 3. Results and Discussion

The behavior of the balloon-expandable stent (Palmaz-Schatz specification [59]) was analyzed using the COMSOL Multiphysics computing package. It should be noted that the purpose of the current study was not to directly simulate a specific design, but rather assess the most dangerous deformation modes significantly affecting the placement of biomedical stents. The FEM method was used to obtain the most unsafe areas of plastic-deformation localization, and to conduct research of these areas with the model, explicitly including the main mechanisms of plastic deformation.

Often, balloon-expandable stents unevenly deform during stent placement. As a result, these deformations can lead to stent damage up to full rupture. The radius of the stent before expansion is 0.75 mm (Figure 2). The simulation used a Cartesian coordinate system with a reference point in the middle of the stent. Inside the stent, a pressure of 1.6 MPa was set along the *x*-axis. Outside the stent, the boundary conditions were free. Thus, the stent expanded up to a radius of 2 mm evenly along its entire length. The stent length changed from 10 to 6.7 mm when expanding along the *x*-axis.

The elastoplastic flow model was used to describe stent deployment; the values of the model parameters for stainless steel are given in Table 1. To build a mesh, three-dimensional simplex elements were used. The minimum and maximum mesh sizes were empirically selected to achieve convergence of the numerical procedure; they were $3\times 10^{-5}$ and $6\times 10^{-4}$ m, respectively.

In the modeling of the stress-strain state of a stent, it was hard to specify the deformation conditions, for which individual parts of the structure could accumulate plastic deformations and collapse at relatively low stresses. Based on the analysis of the stent model, built in the COMSOL Multiphysics software package, the relationship between the values of the principal deformations were determined and the principal deformation directions of the structure parts vulnerable to fracture were found. Considering the coaxiality of the principal stresses and deformation vectors in the elastic region, the value characterizing the strain state type was obtained:(13)$$\mu_{\varepsilon }=2\frac{\varepsilon_{2}-\varepsilon_{3}}{\varepsilon_{1}-\varepsilon_{3}}-1,$$
where $\mu_{\varepsilon }$ is the Lode parameter for the deformed state and $\varepsilon_{1}$, $\varepsilon_{2}$, and $\varepsilon_{3}$ are the principal deformations. The principal strains directions are {0.945, −0.292, 0.147}, {0.194, 0.862, 0.468}, and {−0.263, −0.414, 0.872}.

Figure 3 and Figure 4 show distributions of von Mises stress in the stent before and after expansion, respectively. The maximum von Mises stress value is 407 MPa. The result defined the zones of maximum intensity of plastic deformations and identified the eigenvalues $\varepsilon_{i}$ and the principal direction vectors **k***_i_* of the deformation tensor in pressure. This was necessary to determine the deformed state of the structure and the deformation directions in the principal stress space to use the developed model (1)–(12).

To describe the behavior of the structural material in the plastic deformation process, an elastoviscoplastic two-level model was employed with explicit consideration of the main physical mechanisms and processes at the macro- and mesolevel. The macrolevel polycrystalline aggregate consisted of 350 grains, with uniformly distributed orientations. The grain-size distribution followed the lognormal law; its parameters are discussed in more detail in [44,45,46]. Numerical experiments were carried out for the three-dimensional case, for an anisotropic material with face-centered cubic lattice. The elastic and plastic parameters of model were taken from [47] and corresponded to 316L stainless steel (at room temperature). To obtain the values of hardening parameters and in Equations (10) and (11), a procedure for identifying these parameters based on the experimental data on the tensile strength of steel plates [60] was implemented. The yield strength and initial critical stresses of the steel were determined using the Hall-Petch law by setting an average grain size in the polycrystal [61]. The values of the model parameters are given in Table 2 (unless specified otherwise).

To identify hardening parameters, a numerical experiment of the uniaxial tensile test of the polycrystals was carried out using the hardening laws (10) and (11). A representative volume experienced uniaxial tensile deformation along the x1-axis at the following rate:(14)$$\dot{\varepsilon }={\dot{\varepsilon }}_{0}k_{1}k_{1}-\frac{{\dot{\varepsilon }}_{0}}{2}k_{2}k_{2}-\frac{{\dot{\varepsilon }}_{0}}{2}k_{3}k_{3},$$
where **k***_i_* are the orthogonal unit vectors along the corresponding xi-axes.

The dependence of the Cauchy stress tensor intensity, $\sigma_{u}$, on the small strain tensor intensity, $\varepsilon_{u}$, calculated with numerical simulations (Figure 5) demonstrates a reasonable correspondence with experimental data [60]. It was evident that stress growth occurred nonlinearly, due to mechanisms of intragrain and grain-boundary hardening in the material.

In order to assess the influence of grain size on the deformation behavior of the material, the dependence of the yield strength $\sigma_{0,2}$ on the parameters μ and σ of a lognormal distribution law $\mathrm{LogN}(\mu ,\sigma^{2})$ was suggested (Figure 6). The relationship of the lognormal-law parameters with average grain size is determined as:(15)$$d=e^{\mu +0.5\sigma^{2}}.$$

The range of the law parameters was chosen in such a way so that the grain size in the material changed from 3 to 150 μm. The grain-size distributions in a polycrystal with 350 grains for some parameter values are shown in Figure 7. Thus, an increase in the parameters μ and σ of the lognormal distribution law leads to a significant reduction in the yield strength due to the grain-size growth. It should be noted that the direct effect of the grain size on the material behavior occurred even after reaching the yield strength due to grain boundary hardening.

Using the Lode parameter (13) calculated in the simulation in COMSOL Multiphysics, a relationship between the quantities, $\varepsilon_{1}$, $\varepsilon_{2}$, and $\varepsilon_{3}$, corresponding to the plane in the principal deformation space, $E_{1}E_{2}E_{3}$, was obtained (Figure 8) for the stent part vulnerable to fracture. The material response differed for various deformation directions along this plane. Figure 9 shows a polar diagram for the yield strength depending on the deformation direction in the most dangerous area of the design, shown in Figure 3 and Figure 4.

Each point of the diagram has two polar coordinates:
Angle φ between the deformation direction and the projection of the *X*-axis of principal deformation E_1_ on a given plane;The radius vector drawn from the origin to the considered point, which is equal to the yield strength of the material.

The minimum yield strength, with a value of about 258 MPa, was obtained for the unit vectors, {0.998, 0, −0.062}, {−0.586, 0.537, 0.607}, {−0.998, 0, 0.062}, and {0.586, −0.537, −0.607}, setting the deformation direction in the principal deformation space (other deformation directions give a larger yield strength value). These vectors defined the relationship between the principal deformations and formulated the kinematic loading conditions, at which plastic deformation occurred at the lowest stress in the material. Thus, the most dangerous deformation directions for structure parts with the highest stresses (Figure 3 and Figure 4) in the process of stent expansion were defined.

## 4. Conclusions

The influence of the 316L stainless steel stent microstructure on the mechanical strength and durability in the process of design deployment was studied in this paper. Modeling the process of stent expansion under linearly distributed loading using the COMSOL Multiphysics software package found non-uniform stress-strain fields with the most dangerous zones of design in terms of highest plastic deformation and stresses. To describe the mechanisms and processes at the grain level in the plastic deformation process in these zones, a two-level elastoviscoplastic model was built, allowing to directly control the internal structure of the material, such as grain orientation and grain size in a polycrystal. The numerical simulation based on the developed model showed a good correlation with experimental data. It was found that an increase in the parameters μ and σ of the lognormal distribution law led to the grain-size growth in the polycrystal and, consequently, caused a significant drop in the yield strength. The most dangerous deformation direction for the structure part with the highest stresses and plastic deformation in the process of stent expansion were obtained along the unit vectors {0.998, 0, −0.062}, {−0.586, 0.537, 0.607}, {−0.998, 0, 0.062}, and {0.586, −0.537, −0.607} in the principal deformation space. The yield strength for these deformation directions had a minimum value of about 258 MPa.

## Figures and Tables

**Figure 1 materials-15-06948-f001:**
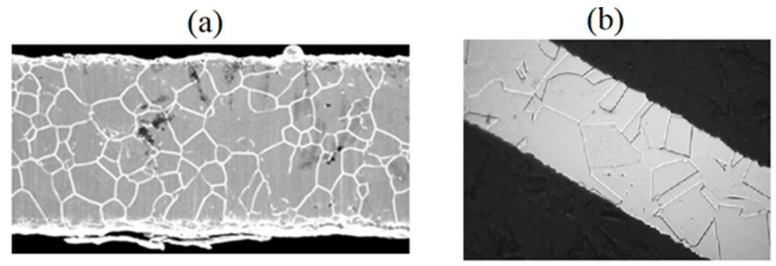
Images of 316L stainless steel stent strut with width of (**a**) 75 [50] and (**b**) 80 [49].

**Figure 2 materials-15-06948-f002:**
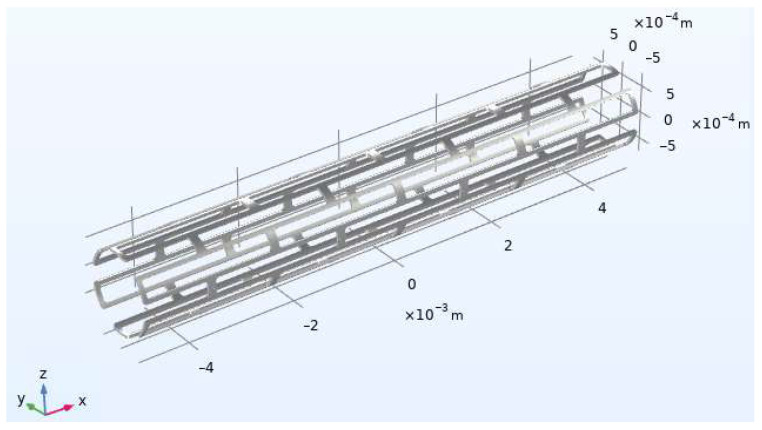
Initial stent configuration (before expansion).

**Figure 3 materials-15-06948-f003:**
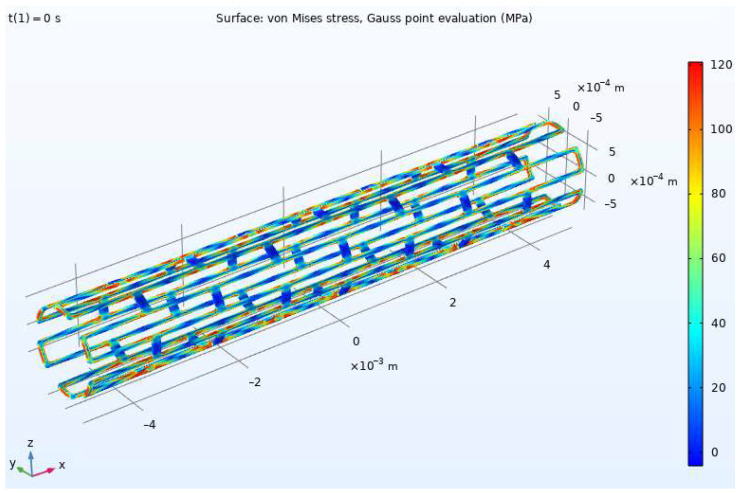
Von Mises stresses for stent before expansion.

**Figure 4 materials-15-06948-f004:**
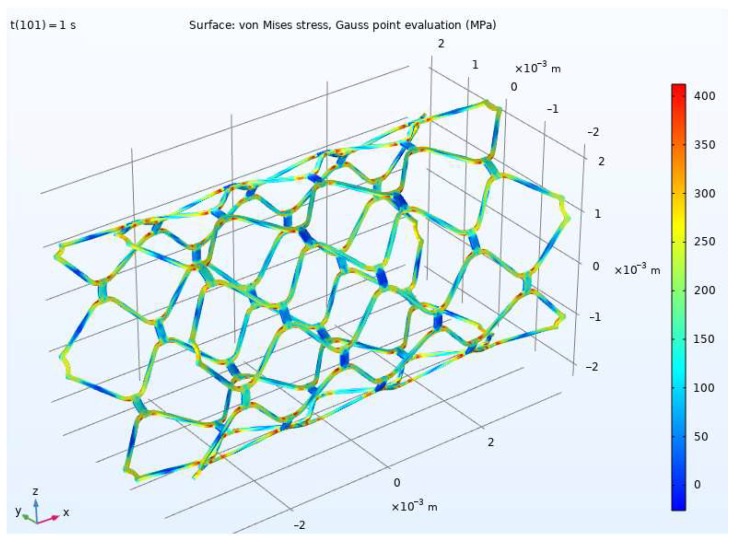
Von Mises stresses for stent after expansion.

**Figure 5 materials-15-06948-f005:**
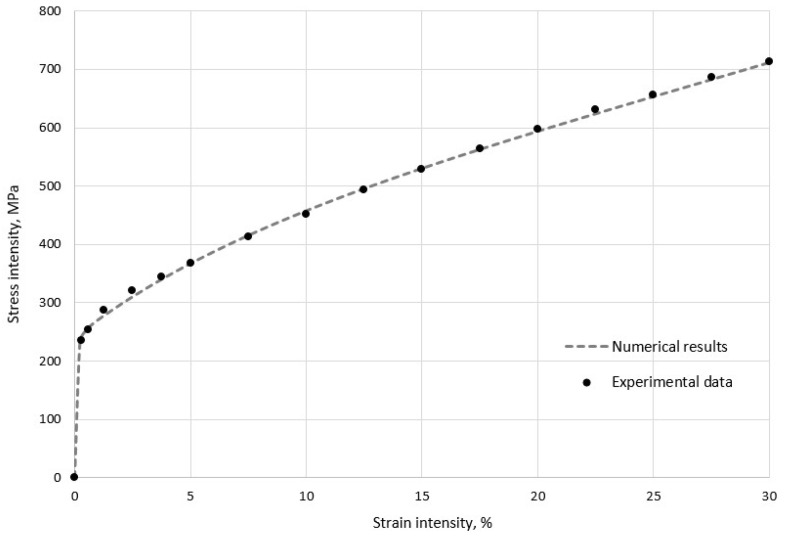
Numerical and experimental results for stress-strain diagram of polycrystalline aggregate in uniaxial tension.

**Figure 6 materials-15-06948-f006:**
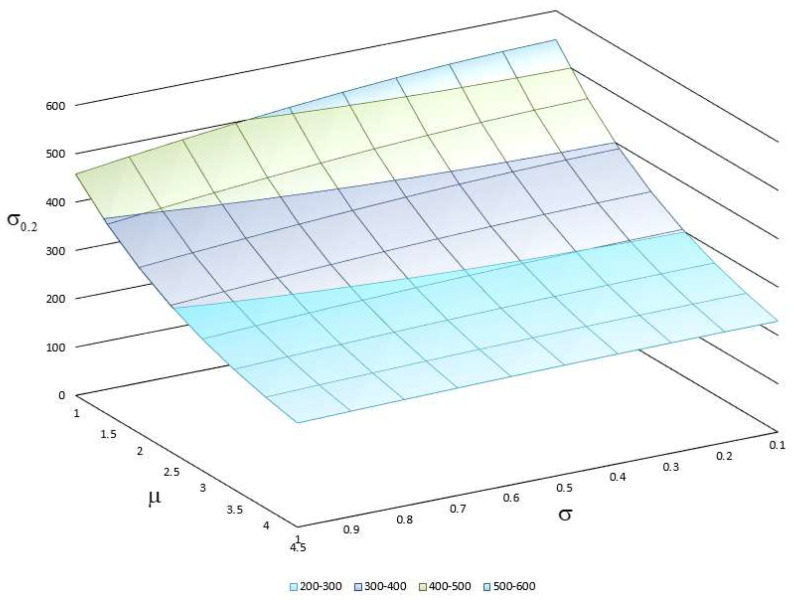
Dependence of yield strength on parameters of lognormal law of grain-size distribution in polycrystal.

**Figure 7 materials-15-06948-f007:**
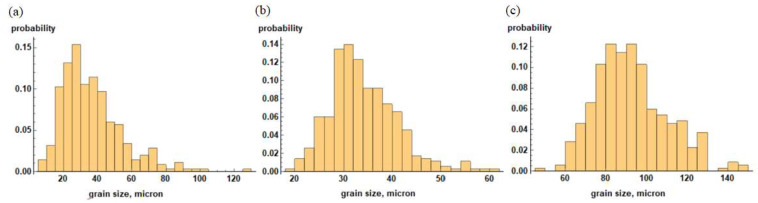
Histograms of grain-size distribution in polycrystal for different parameters: (**a**) μ = 3.5, σ = 0.5; (**b**) μ = 3.5, σ = 0.2; and (**c**) μ = 4.5, σ = 0.5.

**Figure 8 materials-15-06948-f008:**
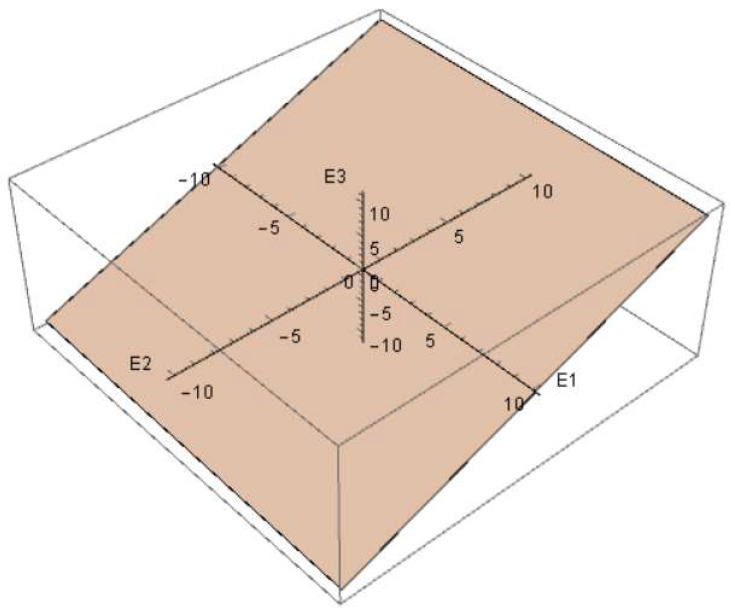
Plane in principal deformation space for the most dangerous area during stent expansion.

**Figure 9 materials-15-06948-f009:**
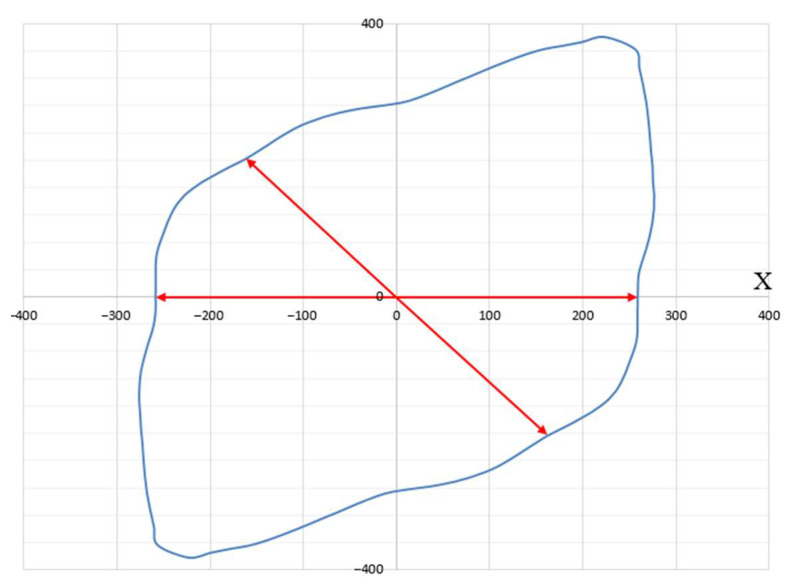
Dependence of yield strength on deformation direction in most dangerous stress state (deformation directions with minimum yield strength are highlighted with red arrows).

**Table 1 materials-15-06948-t001:** Parameters of elastoplastic model for stainless steel.

Material Parameters	Notation	Value
Poisson’s ratio	ν	0.27
Young’s modulus	E	197 GPa
Density	ρ	7000 kg m^−3^
Initial shear stresses	$\sigma_{gs}$	101 MPa

**Table 2 materials-15-06948-t002:** Parameters of elastoplastic model for stainless steel.

Material Parameters	Notation	Value
Grain count	*N*	350
Elastic constant	$C_{iiii},\;i,j=\bar{1,3}$	163 GPa
Elastic constant	$C_{iijj},\;i,j=\bar{1,3}$	110 GPa
Elastic constant	$C_{ijij},\;i,j=\bar{1,3}$	101 GPa

## Data Availability

Not applicable.

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
