# Peer review of "Inelastic Deformation of Coronary Stents: Two-Level Model"

_materials, 2022, doi:10.3390/ma15196948_

Round 1

Reviewer 1 Report

This study describes the internal structure of materials used to produce medical stents by means of two-level elastoviscoplastic mathematical model, which attempts to explain the inelastic deformation of the balloon-expandable stainless steel stent from the microscopic structure. The study concluded that a yield surface for different deformation paths in the principal deformation space for the stent was obtained and the effect of grain size on the deformation behavior of this material was explored using the developed model, which provides a certain material basis for further development of more ideal balloon-expandable stent, regardless of more and more self-expandable stents are in clinic application. But there are still minor questions that need to be solved for publication.

1.     In “Materials and Methods” part, the study did not specify the source and specification of the balloon-expandable stent.

2.     In “Results and Discussion” part, the expansion pressure of the balloon-expandable stent is between the nominal pressure and the burst pressure, and different pressures determine the expansion shape of the stent, as well as other factors such as size of target artery and figure of the diseased artery. Figure 3 and Figure 4 show size of the balloon-expandable stent is from 0.75 mm to 2 mm. If the size of the balloon-expandable stent is increased to 2.1 mm ~ 2.5 mm, What is the result?

Author Response

I sincerely thank you for your work and your questions.
Answers to your questions:

Q1: In “Materials and Methods” part, the study did not specify the source and specification of the balloon-expandable stent.
A1: Added in text (Palmaz-schatz specification of stent)

Q2: In “Results and Discussion” part, the expansion pressure of the balloon-expandable stent is between the nominal pressure and the burst pressure, and different pressures determine the expansion shape of the stent, as well as other factors such as size of target artery and figure of the diseased artery. Figure 3 and Figure 4 show size of the balloon-expandable stent is from 0.75 mm to 2 mm. If the size of the balloon-expandable stent is increased to 2.1 mm ~ 2.5 mm, What is the result?
A2: We tried to expand the stent even more (up to 3 mm), but we limited ourselves 2 mm for reasons of sufficiency to identify dangerous zones. 

Reviewer 2 Report

The paper overall describes an elastic to plastic deformation material model to use in Comsol. It is descriptive in the efforts to show the math models used to describe elastic to plastic behavior of 316L stainless steel.

The discussion section needs improvement on the flow of the paper and the stent simulation needs explanation of their results.

Introduction

Line 41 – Replace “decompose” with degrade and “disintegrate” with resorb. The metal degradable materials are absorbed into the cells and metabolized as well as polymeric materials such a PLLA.

Line 93 – What are the units for grain size?

The paper seems to flow better if the validation parameter used in the stent simulation were stated before it. Line 226 – 288 were before the 3D simulation of the stent in Line 195 - 225.

Results and Discussion

Lines 193 - 221

What is the final diameter the stent is expanded, is it 2.75 mm diameter?

What is the foreshortening of the stent in the simulation?

What are the boundary conditions applied to the stent in the simulation?

What coordinate system is being used in the simulation, i.e. Cartesian, cylindrical?

Why use a direct pressure to inside of stent, most use a rigid balloon to expand the stent or a pressurized balloon to expand the stent?

Why use a high direct pressure inside of stent, most stents are expanded by 12 -14 ATM? This is well below the 200 MPa used in the simulation.

Why was stent model validation not performed such as radial compression of the stent numerically and experimentally? Most stent simulations are validated using radial compression test of actual stents.

Lines 226 – 230 : Discussion of FEM results on stent. There are none, only model validation prior to stenting using uniaxial tensile testing of stainess steel 316L.

There is no description of the maximum von Mises stresses in the simulation results nor the maximum principal strains. Are the stresses in the stent model above 258 MPa considered to be in yielding?

What is the scale for it on Figure 4 meaning if everything above 258 MPa is red, what does the rest of the stent look like on the color scale?

The assumption is the red zones are identified as areas that are prone to fracture or have the highest probability of failure? Please explain.

Author Response

Sincerely thank you for your work and your questions.

Answers to your questions:

Q1: Line 41 – Replace “decompose” with degrade and “disintegrate” with resorb. The metal degradable materials are absorbed into the cells and metabolized as well as polymeric materials such a PLLA.

A1: Added in text (Line 41).

Q2: Line 93 – What are the units for grain size?
A2: Our mistake. Added in text units (μm) (Line 93).

Q3: The paper seems to flow better if the validation parameter used in the stent simulation were stated before it. Line 226 – 288 were before the 3D simulation of the stent in Line 195 - 225.

A3: In this case, we will beat the sequence of presentation. First, we carry out calculations in the COMSOL to determine the zones of maximum intensity of plastic deformation and identify the eigenvalues and the principal direction vectors of the deformation tensor, only then, based on the results of the main vectors of deformation, we calculate the most dangerous directions of deformation with a multilevel model.

Q4: What is the final diameter the stent is expanded, is it 2.75 mm diameter?

A4: We simulated the balloon-expandable stent is from radius 0.75 mm to 2 mm. The final size of the stent is 2 mm.

Q5: What is the foreshortening of the stent in the simulation?
A5: Added in text. The stent length changed from 10 mm to 6.7 mm when expanding along the x-axis (Line 200).

Q6: What are the boundary conditions applied to the stent in the simulation?
A6: Added in text. Inside the stent, pressure of 1.6 MPa was set along the x-axis. Outside the stent, the boundary conditions are free (Lines 198, 199).

Q7: What coordinate system is being used in the simulation, i.e. Cartesian, cylindrical?
A7: Added in text. We used Cartesian coordinate system in the simulation (Line 197).

Q8: Why use a direct pressure to inside of stent, most use a rigid balloon to expand the stent or a pressurized balloon to expand the stent?
A8: The properties of the balloon, in our opinion, do not affect the deformation behavior of the stent. Therefore, it is more important for us what kind of loading program a given cylinder sets. We do not consider the structure and properties of the balloon, believing that to describe the behavior of the stent, it is sufficient to know the loads that the balloon exerts on the stent.

Q9: Why use a high direct pressure inside of stent, most stents are expanded by 12 -14 ATM? This is well below the 200 MPa used in the simulation.

A9: We checked our calculations. The load is 200 MPa is our mistake. The load pressure inside boundary is 1.6 MPa (Line 198).

Q10: Why was stent model validation not performed such as radial compression of the stent numerically and experimentally? Most stent simulations are validated using radial compression test of actual stents.

A10: We do not compare with known experimental data, as this was not the main purpose of the work. The main goal is to obtain dangerous zones.

Q11: There is no description of the maximum von Mises stresses in the simulation results nor the maximum principal strains. Are the stresses in the stent model above 258 MPa considered to be in yielding? What is the scale for it on Figure 4 meaning if everything above 258 MPa is red, what does the rest of the stent look like on the color scale?

A11: Added in text (Lines 217, 220). At different points of the construction, we have a different deformed state. A yield strength of 258 MPa was obtained for the most dangerous zones. It is not a fact that the same deformed state is realized for other points.

Q12: The assumption is the red zones are identified as areas that are prone to fracture or have the highest probability of failure? Please explain.

A12: The red zones are identified areas with high probability of failure. It is not known in advance whether the destruction will occur in these places.